# Nerve Growth Factor, Antimicrobial Peptides and Chemotherapy: Glioblastoma Combination Therapy to Improve Their Efficacy

**DOI:** 10.3390/biomedicines11113009

**Published:** 2023-11-09

**Authors:** Alexandr Chernov, Igor Kudryavtsev, Aleksei Komlev, Diana Alaverdian, Anna Tsapieva, Elvira Galimova, Olga Shamova

**Affiliations:** 1Institute of Experimental Medicine, WCRC “Center for Personalized Medicine”, Saint-Petersburg 197022, Russia; igorek1981@yandex.ru (I.K.); komlev1420@yandex.ru (A.K.); anna.tsapieva@gmail.com (A.T.); oshamova@yandex.ru (O.S.); 2Medical Genetics, Department of Medical Biotechnologies, University of Siena, 53100 Siena, Italy; diana.alaverdian@dbm.unisi.it; 3Sechenov Institute of Evolutionary Physiology and Biochemistry, Russian Academy of Sciences, Saint-Petersburg 194223, Russia; 4Department of Biochemistry, Saint Petersburg State University, Saint-Petersburg 199034, Russia

**Keywords:** glioblastoma, nerve growth factor, antimicrobial peptides, chemotherapy, synergistic effect

## Abstract

Glioblastoma (GBM) is an aggressive and lethal malignancy of the central nervous system with a median survival rate of 15 months. We investigated the combined anticancer effects of nerve growth factor (NGF), cathelicidin (LL-37), and protegrin-1 (PG-1) with chemotherapy (temozolomide, doxorubicin, carboplatin, cisplatin, and etoposide) in the glioblastoma U251 cell line to overcome the limitations of conventional chemotherapy and to guarantee specific treatments to succeed. The MTT (3-(4,5-dimethylthiazol-2-yl)-2,5-diphenyltetrazolium bromide) assay was used to study cell viability and to determine the cytotoxic effects of NGF, LL-37, and PG-1 and their combination with chemotherapy in U251 cells. Synergism or antagonism was determined using the combination index (CI) method. Caspase-3 activity was evaluated spectrophotometrically using a caspase-3 activity assay kit. Apoptosis was analyzed with flow cytometry using propidium iodide (PI) and YO-PRO-1. NGF and the peptides showed a strong cytotoxic effect on U251 glioma cells in the MTT test (IC_50_ 0.0214, 3.1, and 26.1 μM, respectively) compared to chemotherapy. The combination of PG-1 + etoposide had a synergistic effect on apoptosis of U251 glioma cells. It should be noted that the cells were in the early and late stages of apoptosis, respectively, compared with the control cells. The caspase-3 activation analysis revealed that the caspase-3 level was not significantly (*p* > 0.05) increased in U251 cells following PG-1 with etoposide treatment compared with that in the untreated cells, suggesting that the combination of PG-1 and etoposide may induce caspase-independent apoptosis in U251 cells. NGF, LL-37, and PG-1 represent promising drug candidates as the treatment regimen for GBM. Furthermore, the synergistic efficacy of the combined protocol using PG-1 and etoposide may overcome some of the typical limitations of the conventional therapeutic protocols, thus representing a promising approach for GBM therapy.

## 1. Introduction

GBM is the most malignant brain tumor with high mortality in adults [1]. According to the International Agency for Research on Cancer (Globocan), the incidence of brain tumors was assessed in 308,102 cases in 2020 [2]. The incidence of GBM in Caucasians compared to other populations is 2.0 times higher, and in males compared to females, 1.6 times higher.

The treatment for GBM includes maximal safe surgical resection followed by radiation therapy and chemotherapy [3] with the recent addition of tumor-treating fields (TTFields) [4]. Despite the combined modality treatment with surgery, radiotherapy, and chemotherapy, the improvement of the survival rate of glioma patients is still limited. The median survival time for patients is only approximately 15 months, and the 2-year survival rate is only 26.5% [5]. Moreover, the disease rapidly progresses and leads to relapse at 8–9 months post diagnosis [6]. Recent studies demonstrate that tumor heterogeneity with mixtures of genetically distinct subclones may contribute to drug resistance and disease relapse [7,8,9]. Therefore, new and effective therapeutic methods for the successful treatment of GBM are desperately needed.

In addition to ongoing estimations of novel immune-, viro-, and geno-therapies, recent studies of the genome, epigenome, transcriptome, proteome, and metabolome within GBM cancer cells have identified several potential biomarkers and therapeutic targets, including cell cycle and DNA damage repair pathways, angiogenesis inhibitors, and inhibitors of signaling [10,11,12]. Numerous studies demonstrate that NGF and cationic antimicrobial peptides (AMPs) possess significant therapeutic potential and can be considered as candidates for the next generation of anticancer drugs [13,14].

NGF is the first discovered and best characterized member of the neurotrophins (NTs) family, essential for the survival, differentiation, and functional activity of peripheral sensory and sympathetic nerve cells [15,16]. Via the initiation of precursor cells and other growth factors, NGF supports the repair of neurons damaged by ischemia, inflammation, or trauma [17,18]. The evidence obtained from in vitro and in vivo studies shows that NGF inhibits cancer cell proliferation or mitogenesis [19,20,21,22,23,24]. Topical ocular NGF administration is safe and effective, reducing glioma in vivo and the progression of pediatric optic glioma [25,26,27,28]. Further studies might be necessary to elaborate novel in vitro and in vivo techniques to verify the antitumoral activities of NGF.

AMPs, also known as host defense peptides, are part of the innate immune response found in a wide variety of life forms from microorganisms to humans. To date, more than 5000 AMPs have been discovered or synthesized [29]. In vitro studies demonstrate that AMPs possess biological activities towards bacteria, fungi, and some viruses, interacting with negatively charged lipids on cell membranes through electrostatic interactions [29,30,31,32]. In addition to antibacterial activities, AMPs have anticancer activities, suggesting a new strategy for cancer therapy [33,34,35]. The electrostatic interactions between cancer cells and AMPs are crucial for selective binding with cancer cell membranes and selective connection with ion channels [33,34,35]. AMPs are an effective alternative to conventional chemotherapeutics with a low propensity to elicit the development of drug resistance as well as to display toxicity to healthy cells [36,37,38,39]. The AMPs, on the basis of their activities, are divided into those that are toxic to bacteria and cancer cells but not lethal to normal mammalian cells and those that are toxic to bacteria, cancer cells as well as normal mammalian cells [32,40]. The antimicrobial peptides LL-37 and PG-1 are toxic to normal mammalian cells [29,40].

Combination treatments delay the development of drug resistance and reduce the dosage of individual drugs, thereby minimizing side effects [41,42]. Studies on the combinatorial effects of antimicrobial peptides, especially on U251 glioma cells, are missing.

In this study we examine the effects induced by NGF, LL-37, and PG-1 and their potential antitumor activity in the U251 cell line. Moreover, another objective is to investigate the synergistic effect of NGF, LL-37, and PG-1 to find out the most effective anticancer treatment combination. This study serves as an initial attempt to assess the combination effects of NGF, LL-37, and PG-1 with other standard chemotherapies.

## 2. Materials and Methods

### 2.1. Pharmacological Agents, Chemicals, Peptides, and Reagents

Cathelicidin LL-37 was produced via the Fmoc solid phase synthetic approach on a Symphony X peptide synthesizer (Protein Technologies, Tucson, AZ, USA), using standard synthesis protocols. The porcine protegrin-1 PG-1 (SynPep Corporation, Dublin, CA, USA) was kindly provided to us by Prof. R. Lehrer (University of California, Los Angeles, CA, USA). Nerve Growth Factor β Human (Sigma-Aldrich, Saint Louis, MO, USA), gentamicin sulfate (40 mg/mL solution, Shandong Weifang Pharmaceutical Factory Co., Ldt., Shandong, China), Doxorubicin-lans^®^ (intravascular and intravesical solution, 2 mg/mL, vials 10 mg, Veropharm, Russia), Carboplatin-lans^®^ (concentrate for solution for infusion, 10 mg/mL, vials, 50 mg/5 mL, Veropharm, Russia), Temozolomide (temodal^®^ Capsules, 100 mg, Orion Corporation, Tengstrominkatu 8, FIN-20360 Turku, Finland), Cisplatin-lansS^®^ (cincentrate solution for infusion, 0.5 mg/mL, vials 25 mg/50 mL, Veropharm, Russia), and Etoposide ”Ebewe” (concentrate for solution for infusion, 20 mg/mL, 10 mL, Ebewe Pharma, Ges.m.b.H.Nfg.KG, Unterach am Attersee, Austria) were also used.

### 2.2. Cell Culture

The human glioma cell line U251 was obtained from the Institute of Cytology of the Russian Academy of Sciences (St. Petersburg, Russia) and was authenticated with the short tandem repeat assay. The cell culture was cultured in Dulbecco’s modified Eagle medium (DMEM, Sigma-Aldrich, Saint Louis, MO, USA, catalog No. D5796) supplemented with 10% fetal bovine serum (Sigma-Aldrich, Saint Louis, MO, USA, catalog No. F2442) and gentamicin sulfate 10^−4^ g/mL (Shandong Weifang Pharmaceutical Factory Co., Ltd., Shandong, China). The cell culture was maintained under a humid condition of 5% CO_2_ and 95% air at 37 °C [43,44]. An ethics approval for this study was obtained from the Local Ethics Committee at the Institute of Experimental Medicine, Saint-Petersburg on 21 October 2020 (No. 6/20).

### 2.3. Concentrations of Compounds

The U251 cells were treated with NGF at concentrations of 0.23, 0.37, 0.94, 1.88, 3.7, and 7.5 nM; with PG-1 at concentrations of 2, 4, 8, 16, 32, and 64 μM; and with LL-37 at concentrations of 0.5, 1, 2, 4, 8, and 16 μM for the MTT. We used and compared the IC_50_ of NGF, LL-37, and PG-1 with the IC_50_ of the chemotherapy drugs doxorubicin, carboplatin, temozolomide, cisplatin, and etoposide on human U251 glioma cells. Cisplatin was tested at concentrations of 1.66 mM, and 830, 332, 166, 83, and 3.32 μM; carboplatin was tested at concentrations of 26.9, 2.69, and 1.35 mM, and 673, 269, and 134 μM; temozolomide was tested at concentrations of 15.5, 5.15, and 1.55 mM, and 770, 386, and 155 µM; doxorubicin was tested at concentrations of 920, 460, 73.6, 36.8, 18.4, and 7.36 µM; and etoposide was tested at concentrations of 27, 13.5, 6.79, 3.39, 1.69, and 0.85 µM. Accordingly, we used NGF at 3.7 nM, LL-37 at 2 and 3 μM, and PG-1 at 4 and 8 μM concentrations in both the trypan blue exclusion and MTT assays, respectively.

We determined the cytotoxicity index (CI) by using the untreated cells as a negative control. The percentage of cytotoxicity N (%) was calculated with the following Equation (1):(1)$$N(\%)=(\frac{1-sample}{control})\times 100$$
where N% is the CI of the reagents; sample is the cell survival rate treated with chemotherapeutic agents, NGF, LL-37, and PG-1, as well as NGF, LL-37, PG-1, and chemotherapy combinations; and control is the cell survival rate in the control [45].

### 2.4. MTT Assay

We used the MTT assay to evaluate the effects of NGF, LL-37, PG-1, temozolomide, doxorubicin, carboplatin, cisplatin, and etoposide toward the U251 cells [46,47]. U251 glioma cells were seeded in a 96-well plate at 10,000 cells/well and incubated for 24 h at 37 °C with 5% CO_2_. The cells were treated with different chemotherapy drugs and peptides in various concentrations and incubated for a further 24 h. Briefly, 2-fold serial dilutions of NGF, PG-1, LL-37, and 2-fold to 10-fold serial dilutions of the chemotherapy drugs were added to glioma U251 cells in a 96-well cell culture plate. Each assay was performed at least three times and is represented as the mean of the different experiments. The negative controls (0% cell survival) were made with 100 μL of the DMEM, whereas the positive controls were set up with U251 cells + 50 μL of DMEM. A total of 25 µL MTT (5 mg/mL) was added. Subsequently, the MTT (Thiazolyl blue tetrazolium bromide) solution (25 µL, 5 mg/mL) was added to each well, and incubation was allowed to continue for a further 3 h. The absorbance of formazan in each well was measured at 570 nm and 690 nm for reference using a plate reader (Molecular Devices, San Jose, CA, USA).

The percentage (%) of survival cells could be defined with Equation (2) [48]:(2)$$\;Survival\left(\%\right)=\frac{OD\;sample\;-\;OD\;0\%\;survival}{OD\;100\%\;survival\;-\;OD\;0\%\;survival}\;\times 100\%$$

where OD sample, OD 0% survival, and OD 100% survival are the values for the test sample, negative control (0% cell survival), and positive control (100% cell survival), respectively.

### 2.5. Assessment of Drug-Concentration Effect and Calculation of the Combination Index

The cytotoxicity results are expressed as the IC50, the half-maximal (50%) inhibitory concentration of NGF, PG-1, LL-37, and the chemotherapy drugs. The IC50 was calculated using the OriginPro 8.5.1. Software (OriginLab, Northampton, MA, USA) with nonlinear curve fitting. The IC50 values presented (±standard deviation) are the average values obtained from three independent experiments. We calculate the IC50 for the combination of NGF, PG-1, LL-37 and the chemotherapy drugs using the Formula (3):IC50 chemotherapy = IC50 combination × W(3)
where W is the proportion of the chemotherapy drugs in the combination.

The cells were treated with a combination of NGF, PG-1, LL-37, and the chemotherapy drugs to calculate the combination index (CI) according to the Chou–Talalay method [49], using the CompuSyn 2.0 software (ComboSyn, Inc., Paramus, NJ, USA). The CI was evaluated with the Equation (4):(4)$$CI=\frac{(D)1}{(Dx)1}+\frac{\left(D\right)2}{\left(Dx\right)2}$$
where (Dx)1 и (Dx)2 are the doses of substances 1 and 2 used in combination to achieve x % drug effect. D1 and D2 are the doses for single compounds to achieve the same effect. CIs were determined using the unified theory in various doses and mixing ratios (slight synergy CI ˂ 0.85–0.9; moderate synergy CI = 0.7–0.85; synergy CI = 0.3–0.7; strong synergy CI = 0.1–0.3; very strong synergy CI < 0.1; slight antagonism CI < 1.1–1.2; moderate antagonism CI = 1.2–1.45; antagonism CI = 1.45–3.3; strong antagonism CI = 3.3–10; very strong antagonism CI > 10; additivity 0.9 ˂ CI ˂ 1.1).

### 2.6. Assessment of Cell Viability with Flow Cytometry Using YO-PRO-1 and PI

The effects of NGF, PG-1, LL-37, and the chemotherapy drugs on the apoptosis/necrosis of U251 cells were analyzed using flow cytometry. U251 cells at a density of 1 × 10^6^ cells/well were seeded into 6-well plates, incubated for 24 h at 37 °C, and then centrifuged at 1000× *g* for 10 min. Cell viability and cell death machinery were assessed by flow cytometry using YO-PRO-1 vs. PI staining, as it was described in details previously [50,51]. The cells were re-suspended in 200 μL 1× in PBS, and 5 μL YO-PRO-1 and 2.5 μL PI (Sigma-Aldrich, USA, final concentration of PI 1 μg/mL) were added to 100 μL of cell suspension and incubated for 15 min at 37 °C in the dark. The uptake of the dyes was assessed with flow cytometry using a Navios™ flow cytometer (Beckman Coulter, Indianapolis, IN, USA) and Kaluza™ analysis software version 1.2 (Beckman Coulter, Indianapolis, IN, USA).

### 2.7. Caspase Activation Analysis

Caspase-3 activity was evaluated with the «Caspase 3 Assay Kit, Colorimetric» (Sigma, Saint Louis, MO, USA) according to the manufacturer’s protocol [52]. The Caspase 3 Colorimetric Assay Kit is based on the hydrolysis of acetyl-Asp-Glu-Val-Asp p-nitroanilide (Ac-DEVD-pNA) through caspase-3, causing the release of the p-nitroaniline (pNA) moiety. Human glioma U251 cells (5.5 × 10^6^ cells/mL) were plated into 6-well plates and treated with 10 µL etoposide, PG-1, and their combination for 3 h or PBS for the control. Then, the cells were collected and centrifuged for 5 min (4 °C, 600× *g*), re-suspended, and incubated in the lysis buffer on ice for 30 min. The specimens were centrifuged for 20 min (4 °C, 10,000× *g*). The final 100 μL reaction mixture, including 40 μL assay buffer, 50 μL cell lysate supernatant, and the 10 μL Caspase-3 substrate Ac-DEVD-pNA (2 mM), were incubated at 37 °C for 18 h. Caspase-3 activity was assessed at 405 nm using the Multiscan Microplate Readers (ThremoFisher, Waltham, MA, USA).

### 2.8. Statistical Analysis

The experiments were performed at least three times. The Student’s t-test was used to determine the statistical significance of the differences between the means of the different treatments and their respective control groups. The data were calculated with the standard deviation and considered significant at *p* < 0.05. The differences between two independent groups with a small number of samples (*n* < 30) were compared with the nonparametric Mann–Whitney U-test [53]. The descriptive statistics were carried out with the GraphPad Prism software, version 8.01, for Windows (GraphPad Software, La Jolla, CA, USA).

## 3. Results

### 3.1. Sensitivity of U251GBM Cells to NGF, LL-37, PG-1, and Chemotherapy and IC50 Calculation

For determining the cytotoxic effects of NGF, LL-37, PG-1, and chemotherapy, the MTT assay was performed using the U251 cell line. The results showed that the U251 cells were sensitive to treatment in a dose-dependent manner. The dose–response curves are shown in Figure 1. The IC50 values for the NGF, LL-37, PG-1, and chemotherapeutic agents were established. The IC50 values of the tested agents toward the U251 cells are presented in Table 1. The determination of the IC50 for NGF, LL-37, and PG-1 and the IC50 for the anticancer chemotherapeutic agents showed that NGF and the peptides have a strong cytotoxic effect on the U251 glioma cells in the MTT (IC50 = 2.14 × 10^−9^ M, 3.1, and 26.1 μM, respectively) compared to chemotherapy (Table 1). We found that NGF was more effective than LL-37, PG-1, and chemotherapy in reducing the viability of the human glioblastoma U251 cells (Table 1). The data indicated that NGF had good in vitro activity against the U251 cells. According to the guidelines for preclinical drug testing, a compound of a new class is considered cytotoxic at IC50 ≤ 10^−4^ M if its IC50 ≤ IC50 of the reference substance [54]. In the U251 cells, the IC50 of LL-37, NGF, and PG-1 was significantly low than 10^−4^ M, thus demonstrating cytotoxic effects. Additionally, etoposide had a higher efficacy than the other chemotherapeutic agents. The results showed that etoposide inhibited the proliferation of glioma cells with IC50 values of 4.9 to 25.9 µM.

### 3.2. The Combination Index (CI) and Anti-Tumor Activity of Combined Treatment of NGF, LL-37, and PG-1 with Chemotherapy

We also examined the response of the U251 cell line to 15 different combinations of NGF, LL-37, PG-1, and chemotherapy. The cytotoxicity of the combined treatment of NGF, LL-37, and PG-1 with chemotherapy was measured with the MTT assay. The following drug combinations were used: NGF + chemotherapy (doxorubicin, carboplatin, temozolomide, cisplatin, etoposide), PG1 + chemotherapy (doxorubicin, carboplatin, temozolomide, cisplatin, etoposide), LL37 + chemotherapy (doxorubicin, carboplatin, temozolomide, cisplatin, etoposide), PG-1 + LL-37, PG-1 + NGF, and PG-1 + LL-37 + NGF.

The IC50 values of all the combinations are shown in Table 1. After treatment for 24 h, the corresponding IC50 values (measured using the MTT assay) of the PG-1 combinations with doxorubicin, carboplatin, cisplatin, and etoposide; NGF combinations with doxorubicin, carboplatin, cisplatin, and etoposide; and LL-37 combinations with cisplatin and etoposide were lower than that of the single-drug chemo treatment (Table 1). The data indicated that the presence of PG-1, NGF, and LL-37 could further promote the cytotoxicity of chemotherapeutic agents.

Further, the combinations of PG-1 with carboplatin, temozolomide, cisplatin, and etoposide and combination of NGF with cisplatin exhibited higher toxicity than that of the single-drug chemo treatment (Table 1). The CI values reflect the interaction between the two drugs. Synergism or antagonism was determined using the combination index method. The CI values of the tested agents toward the U251 cells are presented in Table 2. PG-1 combined with etoposide displayed synergistic effects with a CI value of 0.65 on the U251 glioma cells in the MTT, indicating that the combination of PG-1 with etoposide might also be promising.

PG-1 combined with doxorubicin displayed an additive effect with a CI value of 0.94 in the U251 glioma cells with the MTT. Meanwhile, antagonism was found for the other combinations in the U251 cells.

The cytotoxic activities of all possible combinations of the three peptides, NGF, PG-1, and LL-37, were also analyzed on the U251 cultures using the MTT assay (Figure 2). The cytotoxic activity of the combinations of peptides PG-1 + LL-37, PG-1 + NGF, and PG-1 + LL-37 + NGF was significantly (*p*  <  0.05) lower than that of PG-1 (Figure 2). The cytotoxic activity of the combinations PG-1 + LL-37, LL-37 + NGF, and PG-1 + LL-37 + NGF was significantly (*p*  <  0.05) lower than that of LL-37 (Figure 2). The cytotoxic activity of the combinations PG-1 + NGF, LL-37 + NGF, and PG-1 + LL-37 + NGF was significantly (*p*  <  0.05) lower than that of NGF (Figure 2).

### 3.3. Cell Apoptosis Detected with Flow Cytometry

To verify the synergistic effects of PG-1 + etoposide on the U251 cells, the apoptotic effects of PG-1 + etoposide were tested using the YO-PRO-1 and PI apoptosis kit. The representative dot plots illustrating apoptotic status are shown in Figure 3. The percentage of apoptotic cells (early apoptotic plus late apoptotic cells) treated with the PG-1 + etoposide combination (Figure 3D) solution was 70.4 ± 11.4% (early apoptotic, *p* < 0.0001) and 11.7 ± 4.7% (late apoptotic, *p* < 0.0054), which was significantly higher in comparison with etoposide (50.7 ± 5.2%, *p* < 0.0001, 6.3 ± 2.0%, *p* ˃ 0.05) and PG-1 (53.1 ± 2.8%, *p* < 0.0001), respectively. Thus, the combination of PG-1 with etoposide has a synergistic cytotoxic effect in U251 glioma cells by increasing (*p* < 0.0001) the ratio of cells at the stage of early apoptosis compared to the action of etoposide and PG-1. These results indicated the potential synergistic enhancement of cancer therapy using PG-1 + etoposide.

### 3.4. Activation of Caspase-3 on U251 Cells

Caspase-3, one of the critical mediators of apoptosis, performs a key role in regulating both mitochondrial and death receptor apoptotic pathways. The activation of caspase-3 of the U251 cells was evaluated with the Caspase-3 Activity Assay Kit to confirm the probable pathways of the synergistic apoptotic effect of PG-1 + etoposide. The caspase-3 activation analysis demonstrated that the caspase-3 level was not significantly (*p* > 0.05) increased in the U251 cells following PG-1 + etoposide treatment compared with that in the untreated cells, suggesting that the combination of PG-1 and etoposide may induce caspase-independent apoptosis in U251 cells (Figure 4).

## 4. Discussion

Our results showed that NGF, LL-37, and PG-1 were significantly cytotoxic to the U251 GBM cells compared to chemotherapy. We observed a dose-dependent cytotoxic effect with the NGF treatment. The higher doses of NGF (100–200 ng/mL) showed a cytotoxic effect to the U251 GBM cells, whereas low (10, 25, 50 ng/mL) concentrations of NGF induced a mitogenic effect. Previously, we also showed that NGF induced a strong cytotoxic effect on C6 glioma cells in the MTT test and xCELLigence real-time cell analysis [55]. The IC50 values of the combinations of NGF with doxorubicin, carboplatin, and cisplatin were lower than the IC50 values of the chemotherapy drugs alone in the MTT assay. These results indicate stronger cytotoxic antitumor effects of these combinations. Probably, the cytotoxic effects of NGF in glioma U251 cells may be related to its capacity to inhibit the basal oxygen consumption rate, ATP-synthetase, maximal respiration of mitochondria, and migration of these cells [56]. NGF, through interaction with its specific receptors, the p75 neurotrophin receptor (p75NTR) and the tropomyosin-related kinase A (TrkA), regulates carcinogenesis by either suppressing or supporting tumor growth. In vitro and in vivo data demonstrate that NGF inhibits cancer cell proliferation or mitogenesis [19,20,21,22,23,24].

However, other studies propose that NGF stimulates glioblastoma proliferation [50,51,52]. Giraud et al. presented evidence that detection of the p75NTR receptor in the Golgi apparatus could be correlated with a decrease in cell apoptosis that causes U-87 MG human glioblastoma cells to become tumorous [57]. NGF leads to clonal growth of the human glioblastoma cell line via binding of NGF to tyrosine kinase [58]. Singer et al. revealed that NGF, acting via Trk receptor phosphorylation, induced the growth of U251, U87, and U373 cells in culture by 9%, 16%, and 33%, respectively, compared with controls after 3 days [59]. Colocalizations of NGF with gamma-tubulin at the centrosomes throughout the cell cycle and phosphorylated TrkA with alpha-tubulin at the mitotic spindle were detected in a study of the direct mitogenic effects of NGF and TrkA in the glioma cell line U251 [23]. Modulation of the mitosis of human glioma cells by NGF is carried out through phosphorylation of TrkA and tubulin [59]. An analysis of NGF expression in astrocytoma samples from 70 adult patients revealed the overexpression of NGF in astrocytomas compared with that in the control cohort (*p* < 0.05), particularly in grade III (*p* < 0.05) [60]. An association of NGF overexpression in astrocytomas with the generation, location, progression, and pathological grade of the astrocytoma was detected [60]. Yang et al. demonstrated an overexpression of NGF and TrkA in U251 and distribution of NGF in the plasma and nuclei, while TrkA was distributed in the membrane and nucleoli [61]. Johnston et al. found that the expression of p75NTP significantly induced the migration and invasion of genetically distinct glioma cells [62]. In total, inconsistent results in the studies regarding the effects of NGF on U251 glioma cells can be due to the pleiotropic effects and colocalization of TrkA and p75NTR receptors and receptor complexes formation. Interaction of p75NTR with TrkA increased the affinity and selectivity of NGF binding, promoting TrkA signaling, supporting survival, and inducing differentiation of sympathetic neurons [63].

In the current study, we show that the IC50 of the combination of LL-37 with etoposide was lower than the IC50 of the chemotherapy drug alone, while the IC50 of the combinations of LL-37 with temozolomide, cisplatin, carboplatin, and doxorubicin was higher than the IC50 of the chemotherapy drugs. However, the combination of LL-37 with cisplatin in several doses (830, 332, 166, 83, 3.32 μM) and the combination of LL-37 with carboplatin at 2.69 mM had a more cytotoxic effect than the chemotherapy drugs alone. We have recently detected that the cytotoxic effects of LL-37 with temozolomide in glioma U251 cells can be associated with their capacity to inhibit clonogenicity, migration, basal oxygen consumption rate, ATP-synthetase, and maximal respiration of mitochondria in these cells [56].

LL-37 induces tissue-specific or cell-specific effects by interacting with different membrane receptors in various cancer cells. Multiple mechanisms were discovered in lung, breast, gastric, and colon cancers by extensive LL-37 studies [64]. It was found that LL-37 plays a dual role in carcinogenesis, exerting pro- and anti-tumorigenic effects in different cancers [65]. LL-37 regulates tumorigenesis by either suppressing the growth of acute myeloid [66], lymphocytic leukemia [67], and gastric cancer [68] or supporting the growth of breast [69], ovarian [70], and lung cancers [71]. A study investigating LL-37 expression and function in colon cancer showed LL-37 downregulation in colon cancer tissues through the caspase-independent apoptotic pathway [72]. A comparative analysis of the effect of human LL37 and cationic peptides encoded by different viruses on the survival of human U87G glioblastoma cells revealed that LL-37 as well as LL17–32 inhibit the viability of U87G cells in a dose-dependent manner [73]. Whereas U87G is a highly radio-resistant GMB cell line, Colle et al. concluded that LL-37 and LL17–32 may be helpful in the therapy of glioblastomas [73]. An expression analysis of 84 genes related to DNA damage in wild-type and LL-37-knockdown melanoma cells (A375) and in breast cancer (SKBR3) established the downregulation of genes related to stemness, including telomerase reverse transcriptase, forkhead box D3, and undifferentiated embryonic cell transcription factor 1 [74]. Thus, LL37 regulates cancer cell stemness, potentially promoting a high resistance to radiation and chemotherapy.

The IC50 of the combinations of PG-1 with doxorubicin, carboplatin, cisplatin, and etoposide was lower than the IC50 of the chemotherapy drugs alone, which indicates higher cytotoxic effects of these combinations in glioma U251 cells compared with those of the chemotherapy drugs alone. In contrast, the effects of the combinations of NGF, LL-37, and PG-1 with temozolomide were less cytotoxic than the effects of the chemotherapy drug alone. Furthermore, the combinations of LL-37, PG-1, and NGF with temozolomide have less cytotoxic effects in comparison with those of temozolomide in C6 glioma cells [55]. In this study, we revealed that the combination of PG-1 with etoposide indicates a synergy cytotoxic effect in glioma U251 cells. Furthermore, the combination of PG-1 with doxorubicin has an additive cytotoxic effect in glioma U251 cells. Other combinations showed antagonism in these cells.

Porcine protegrin-1 (PG-1) is a potent AMP with broad-spectrum antibacterial and antiviral activity. Amphipathic β-sheets are used to penetrate and form pores in the membrane by several toxins that have been designated [75,76,77,78,79]. Several studies have shown multiple individual mechanisms or a combination of mechanisms of AMPs cytotoxic activities that cause cell lysis, leading to necrosis or programmed cell death [80,81,82]. Soundrarajan et al. estimated PG-1 cytotoxicity to embryonic fibroblasts, retinal cells, embryonic kidney cells, neuroblastoma cells, alveolar macrophage cells, and neutrophils [83]. The study demonstrated that the cellular toxicity of PG-1 depends on its conformational change when binding with various types of cells [83].

Moreover, the current study revealed that the combination of PG-1 and etoposide had a synergistic effect on the apoptosis of U251 glioma cells. The caspase-3 activation analysis demonstrated that the caspase-3 level was not significantly (*p* > 0.05) increased in the U251 cells following PG-1 + etoposide treatment compared with that in the untreated cells, suggesting that the combination of PG-1 and etoposide may induce caspase-independent apoptosis in U251 cells. Ren et al. revealed that LL-37 induces caspase-independent apoptosis in human colon cancer cells through the initiation of the Gi-coupled GPCR-p53-Bcl-2/Bax/Bak-AIF/EndoG pathway [72]. These data support further research to synthesize the LL-37 peptide as a potential inducer of caspase-independent apoptosis.

## 5. Conclusions

Our results showed that NGF, LL-37, and PG-1 were significantly cytotoxic to U251 GBM cells compared to chemotherapy. NGF, LL-37, and PG-1 represent promising drug candidates to offer to patients with GBM. Furthermore, the synergistic efficacy of the combined protocol using PG-1 and etoposide might open new horizons in GBM therapy for its ability to eliminate the typical constraints associated with conventional treatment modalities. However, this current concept should be considered as preliminary, and the results need to be confirmed in future studies.

## Figures and Tables

**Figure 1 biomedicines-11-03009-f001:**
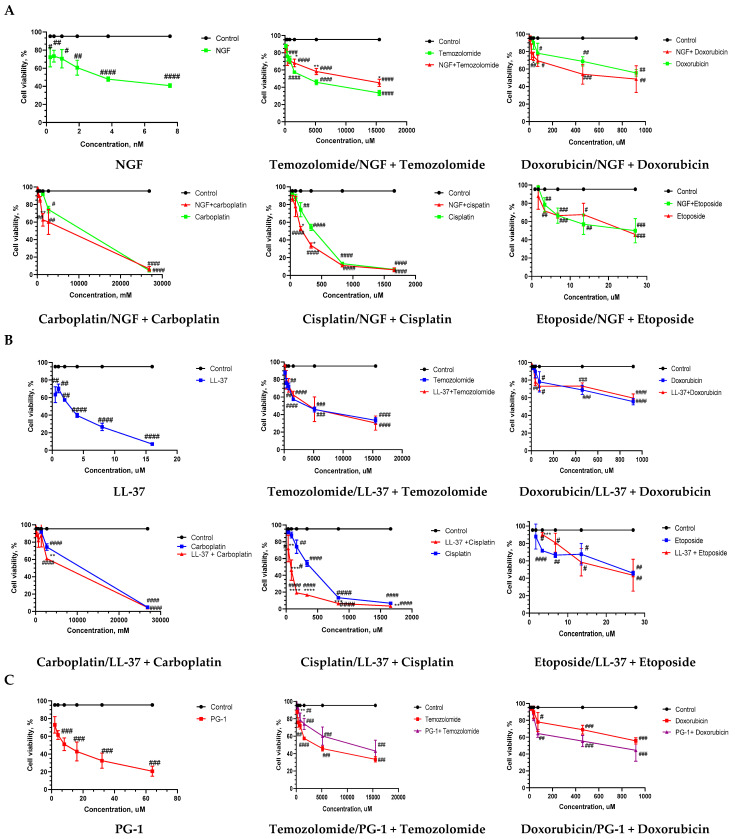
Graph of viability versus drug concentration. Cell viability was assessed with the MTT assay. Dose-dependent effect of NGF (**A**), LL-37 (**B**), and PG-1 (**C**) and their combination with chemotherapy on cell viability at 24 h time-course. Data shown are representative of three separate experiments, and values are given as mean ± SD. Statistically significant difference * *p* < 0.05, ** *p* < 0.01, *** *p* < 0.001, **** *p* < 0.0001 combination effects from drug alone; # *p* < 0.05, ## *p* < 0.01, ### *p* < 0.001, #### *p* < 0.0001—statistically significant difference drug or combination from control.

**Figure 2 biomedicines-11-03009-f002:**
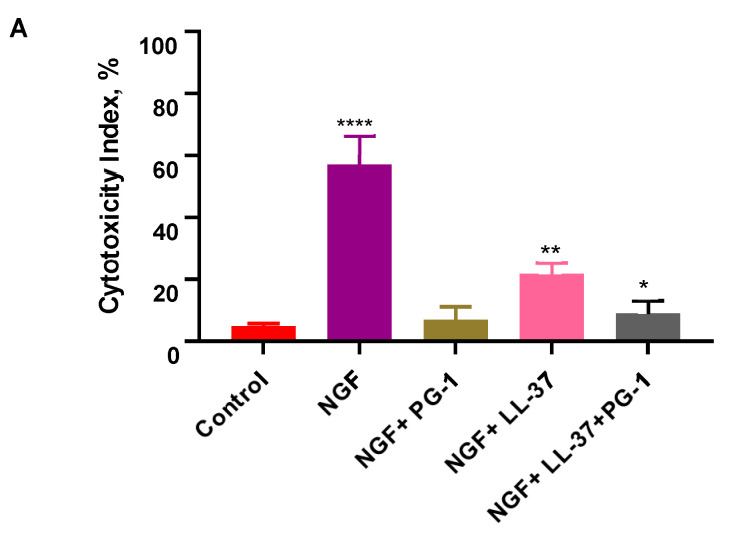
Computed CI values for the combination of NGF on U251 human glioma cells according to the results of the MTT test (**A**); LL-37 with NGF and PG-1 on U251 human glioma cells according to the results of the MTT test (**B**); PG-1 with LL-37 and NGF on U251 human glioma cells according to the results of the MTT test (**C**). The results are presented as the means (columns) ± S.D. (bars) (n = 3, in triplicate). * *p* < 0.05, ** *p* < 0.01, **** *p* < 0.0001 statistically significant differences between CI value of drug or its combinations and control.

**Figure 3 biomedicines-11-03009-f003:**
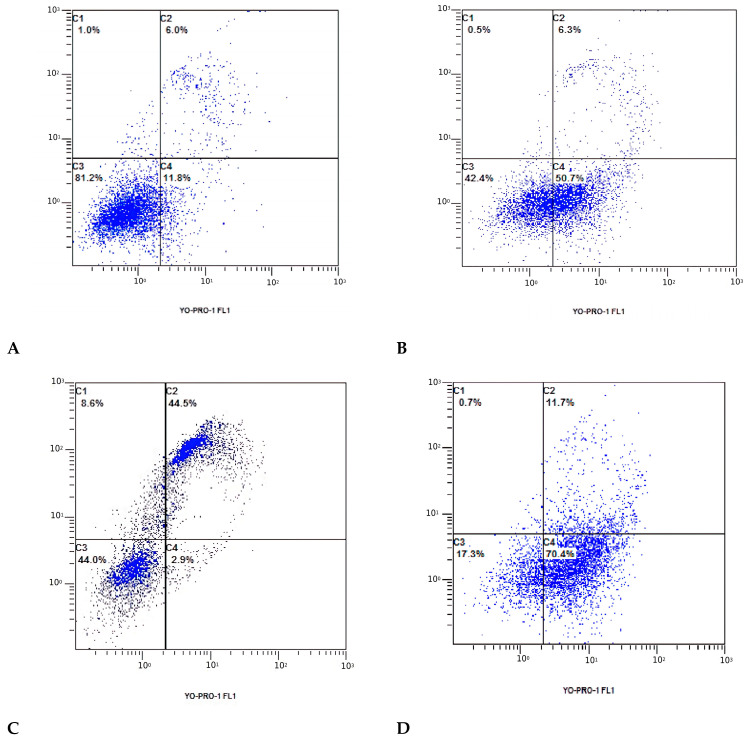
Apoptosis, necrosis, and necroptosis analysis with flow cytometry using PI and YO-PRO-1. U251 cells were treated with etoposide (**B**) and PG-1 (**C**) individually and in combination (**D**) (etoposide + PG-1) for 24 h compared with that in the normal controls (**A**). C3 represents the rate of viable cells; C1 and C2 represent the rate of late apoptosis or necrosis; C4 represents the rate of early apoptosis.

**Figure 4 biomedicines-11-03009-f004:**
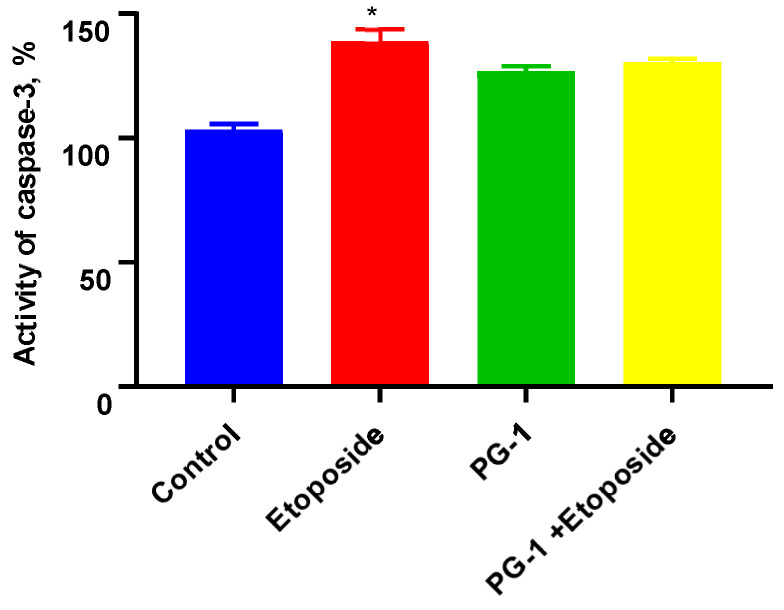
Caspase-3 activity of U251 cells; Data shown are representative of three separate experiments, and values are given as mean ± SD. * *p* < 0.05, statistically significant difference between control and PG-1.

**Table 1 biomedicines-11-03009-t001:** Cytotoxic activity of the combined treatment NGF, LL-37, and PG-1 with chemotherapy in glioma cells (IC50: µM). IC50 values (µM) were obtained with the MTT test.

Compounds	N	Monotherapy, IC50 µM	NGF + Chemotherapy, IC50 µM	LL37 + Chemotherapy, IC50 µM	PG1 + Chemotherapy, IC50 µM
MTT test					
Doxorubicin	3	1554.98 ± 207.5	600.5 ± 55.75	5265.5 ± 1031.0	730.9 ± 12.43
Carboplatin	3	3652.9 ± 670.8	2880.6 ± 275.5	3513.3 ± 493.0	2938.9 ± 529.7
Temozolomide	3	1725.7 ± 494.0	16,804.0 ± 937.4	4007.0 ± 365.5	9761.7 ± 997.0
Cisplatin	3	371.5 ± 23.50	207.4 ± 8.477	869 ± 107.5	237.9 ± 29.83
Etoposide	3	25.90 ± 0.61	13.7 ± 0.1129	17.9 ± 3.03	17.0 ± 1.80
NGF (nM)	3	2.14 ± 5.0			
LL-37	3	3.1 ± 0.4063			
PG-1	3	26.1 ± 7.6			

**Table 2 biomedicines-11-03009-t002:** CI of the combined one-day exposure of NGF, LL-37, and PG-1 with chemotherapy on U251 glioma cells according to the MTT assay. The mean combination index (CI) value of combination treatments in U251 was calculated as explained in the Methods.

Compounds	PG1 + Chemotherapy	LL37 + Chemotherapy	NGF + Chemotherapy
MTT test			
Doxorubicin	0.94 additivity	188.6 very strong antagonism	2.42 antagonism
Carboplatin	1.46 antagonism	4.31 strong antagonism	2.69 antagonism
Temozolomide	6.04 strong antagonism	8.95 strong antagonism	9.73 strong antagonism
Cisplatin	1.38 antagonism	2.67 antagonism	2.78 antagonism
Etoposide	0.65 synergy	1.85 antagonism	2.34 antagonism

## Data Availability

All source data supporting the findings of this manuscript are available from the corresponding authors upon request.

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
