# Peer review of "Nerve Growth Factor, Antimicrobial Peptides and Chemotherapy: Glioblastoma Combination Therapy to Improve Their Efficacy"

_biomedicines, 2023, doi:10.3390/biomedicines11113009_

Round 1
Reviewer 1 Report
In this paper, the authors investigate the combined anticancer effects of NGF, LL-37 and PG-1 in combined with chemotherapy drugs (temozolomide, doxorubicin, carboplatin, cisplatin, and etoposide) in the U251 glioblastoma cell line and used MTT assay to assess cell viability and cytotoxic effects and caspase-3 activity was measured to indicate apoptosis.
However, it's important to note that all the studies conducted so far have been based on in vitro assays, which may not be fully representative of the real physiological conditions. Given the heterogeneity of BBB and the microenvironment in GBM, it is strongly advisable to conduct in vivo studies for a more comprehensive understanding of these therapeutic approaches.
Authors need to enlarge Figure 1 images.
Moderate edits
Author Response
Reviewer 1
Journal Biomedicines (ISSN 2227-9059)
Manuscript ID biomedicines-2632674
Type Article
Title Nerve Growth Factor, Antimicrobial Peptides and Chemotherapy: Glioblastoma Combination Therapy to Improve their Efficacy
Authors Alexandr Chernov *, Igor Kudryavtsev, Aleksei Komlev, Diana Alaverdian, Anna Tsapieva, Elvra Galimova*, Olga Shamova
Section Neurobiology and Neurologic Disease
Special Issue 10th Anniversary of Biomedicines—Novel Targets for Cranial Tumors
Abstract
Glioblastoma (GBM) is the aggressive and lethal malignancy of the central nervous system with a median survival rate of 15 months. We investigated the combined anticancer effects of nerve growth factor (NGF), cathelicidin (LL-37), protegrin-1 (PG-1), with chemotherapy (temozolomide, doxorubicin, carboplatin, cisplatin, and etoposide) in glioblastoma U251 cell line to overcome limitations of conventional chemotherapy and to guarantee specific treatments to succeed. The MTT (3-(4,5-dimethylthiazol-2-yl)-2,5-diphenyltetrazolium bromide) assay was used to study cell viability and to determine cytotoxic effects of NGF, LL-37, PG-1 and their combination with chemotherapy in U251 cells. Synergism or antagonism was determined using combination index (CI) method. Caspase-3 activity was evaluated spectrophotometrically using a caspase-3 activity assay kit. The apoptosis analyzed by flow cytometry using propidium iodide (PI) and YO-PRO-1. NGF and peptides showed a strong cytotoxic effect on U251 glioma cells in the MTT test (IC50 0.0214, 3.1, 26.1 μM, respectively) compared to chemotherapy. The combination of PG-1+etoposide had a synergistic effect on apoptosis of U251 glioma cells. It should be noted that cells were in the early and late stages of apoptosis, respectively, compared with control cells. The caspase 3 activation analysis revealed that the caspase-3 level was not significantly (p>0.05) increased in U251 cells following PG-1 with etoposide treatment compared with those in the untreated cells, suggesting that the combination of PG-1 and etoposide may induce caspase-independent apoptosis in U251 cells. NGF, LL-37, and PG-1 represent promising drugs candidate as the treatment regimen for GBM. Furthermore, the synergistic efficacy of the combined protocol using PG-1 and etoposide may overcome some typical limitations of conventional therapeutic protocols, thus representing a promising approach for GBM therapy.
Dear Reviewer 1,
We are grateful to the reviewers for their thoughtful comments that helped us improve the quality of our manuscript. We answered all the reviewer's comments and modified the manuscript accordingly. We made the following changes according to the reviewer's comments:
Comments and Suggestions for Authors
In this paper, the authors investigate the combined anticancer effects of NGF, LL-37 and PG-1 in combined with chemotherapy drugs (temozolomide, doxorubicin, carboplatin, cisplatin, and etoposide) in the U251 glioblastoma cell line and used MTT assay to assess cell viability and cytotoxic effects and caspase-3 activity was measured to indicate apoptosis.
However, it's important to note that all the studies conducted so far have been based on in vitro assays, which may not be fully representative of the real physiological conditions. Given the heterogeneity of BBB and the microenvironment in GBM, it is strongly advisable to conduct in vivo studies for a more comprehensive understanding of these therapeutic approaches.
Authors need to enlarge Figure 1 images.
Author's Reply to the Review Report (Reviewer 1)
- Moderate editing of English language required
Response: Thank you for your suggestion. Extensive editing of English language was performed by Dr. Darya Gaysina, School of Psychology, University of Sussex, Brighton, UK.
- «…it is strongly advisable to conduct in vivo studies for a more comprehensive understanding of these therapeutic approaches»
Response: Thank you for your constructive comment. We rely on standards of the global academic and professional community in research. We performed in vivo experiments and included these results of our study in other publication - Alexandr N. Chernov, Alexandr V. Kim, Sofia S. Skliar, Evgeniy V. Fedorov, Anna N. Tsapieva, Aleksei L. Сhutko, Marina V. Matsko, Elvira. S. Galimova, Olga V. Shamova. Human cathelicidin LL-37, protegrin PG-1 and their combinations with chemotherapy show new anti-cancer activities through the expression of molecular markers in glioblastoma multiforme cells. The manuscript is under review at the journal's «Cancer Chemotherapy and Pharmacology» editorial office.
This study serves as an initial attempt to assess the combination effects of NGF, LL-37, and PG-1 with other standard chemotherapies. We indicated in Conclusion that current concept (NGF, LL-37 and PG-1 can be promising drugs for the treatment of GBM and can overcome the limitations of conventional therapies) should be considered as preliminary and the results need to be confirmed in future studies.
- Authors need to enlarge Figure 1 images.
Response: Thank you, we corrected it.
We thank you very much for your interest and I look forward to hearing from you.
Sincerely
Authors
Reviewer 2 Report
Chernov et al. investigate the role of utilization of nerve growth factor and antimicrobial peptides in the treatment of glioblastoma. Hypothesis is that the addition of nerve growth factors and cationic antimicrobial peptides have potential therapeutic benefit and may be a candidate for future antineoplastic therapies. The study designed utilized a glioma cell line with the application of nerve growth factor in utilized nerve growth factor, cathelicidin and protegrin-1 along with chemotherapy using a variety of agents.
Abstract: Adequately summarizes the contents of the submission.
Introduction: Adequately presents the background, and the hypothesis to be investigated in the rationale.
Materials and methods: Adequately details the methods utilized in data analysis.
Results: Well written and coherently describes the findings and statistical analysis.
Discussion: well written, explaining in a brief and understandable manner explanations or potential explanations for the results and how these would be applied to future studies.
Conclusion: Adequately summarizes the major findings and how these findings may be applied to future investigations.
Graphs and figures: All are essential and do not duplicate the text of the results section.
References: All are appropriate to the subject matter.
Author Response
Reviewer 2
Journal Biomedicines (ISSN 2227-9059)
Manuscript ID biomedicines-2632674
Type Article
Title Nerve Growth Factor, Antimicrobial Peptides and Chemotherapy: Glioblastoma Combination Therapy to Improve their Efficacy
Authors Alexandr Chernov *, Igor Kudryavtsev, Aleksei Komlev, Diana Alaverdian, Anna Tsapieva, Elvra Galimova*, Olga Shamova
Section Neurobiology and Neurologic Disease
Special Issue 10th Anniversary of Biomedicines—Novel Targets for Cranial Tumors
Abstract
Glioblastoma (GBM) is the aggressive and lethal malignancy of the central nervous system with a median survival rate of 15 months. We investigated the combined anticancer effects of nerve growth factor (NGF), cathelicidin (LL-37), protegrin-1 (PG-1), with chemotherapy (temozolomide, doxorubicin, carboplatin, cisplatin, and etoposide) in glioblastoma U251 cell line to overcome limitations of conventional chemotherapy and to guarantee specific treatments to succeed. The MTT (3-(4,5-dimethylthiazol-2-yl)-2,5-diphenyltetrazolium bromide) assay was used to study cell viability and to determine cytotoxic effects of NGF, LL-37, PG-1 and their combination with chemotherapy in U251 cells. Synergism or antagonism was determined using combination index (CI) method. Caspase-3 activity was evaluated spectrophotometrically using a caspase-3 activity assay kit. The apoptosis analyzed by flow cytometry using propidium iodide (PI) and YO-PRO-1. NGF and peptides showed a strong cytotoxic effect on U251 glioma cells in the MTT test (IC50 0.0214, 3.1, 26.1 μM, respectively) compared to chemotherapy. The combination of PG-1+etoposide had a synergistic effect on apoptosis of U251 glioma cells. It should be noted that cells were in the early and late stages of apoptosis, respectively, compared with control cells. The caspase 3 activation analysis revealed that the caspase-3 level was not significantly (p>0.05) increased in U251 cells following PG-1 with etoposide treatment compared with those in the untreated cells, suggesting that the combination of PG-1 and etoposide may induce caspase-independent apoptosis in U251 cells. NGF, LL-37, and PG-1 represent promising drugs candidate as the treatment regimen for GBM. Furthermore, the synergistic efficacy of the combined protocol using PG-1 and etoposide may overcome some typical limitations of conventional therapeutic protocols, thus representing a promising approach for GBM therapy.
Dear Reviewer 2,
We are grateful to the reviewers for their thoughtful comments that helped us improve the quality of our manuscript. We answered all the reviewer's comments and modified the manuscript accordingly.
We thank you very much for your interest and I look forward to hearing from you.
Sincerely
Authors